# Development of Anticancer Peptides Using Artificial Intelligence and Combinational Therapy for Cancer Therapeutics

**DOI:** 10.3390/pharmaceutics14050997

**Published:** 2022-05-06

**Authors:** Ji Su Hwang, Seok Gi Kim, Tae Hwan Shin, Yong Eun Jang, Do Hyeon Kwon, Gwang Lee

**Affiliations:** 1Department of Molecular Science and Technology, Ajou University, 206 World Cup-ro, Suwon 16499, Korea; js3004@ajou.ac.kr (J.S.H.); rlatjrrl9977@ajou.ac.kr (S.G.K.); jye120@ajou.ac.kr (Y.E.J.); 2Department of Physiology, Ajou University School of Medicine, 206 World Cup-ro, Suwon 16499, Korea; catholicon@ajou.ac.kr (T.H.S.); dohyeon248@ajou.ac.kr (D.H.K.)

**Keywords:** anticancer peptides, cancer therapy, deep learning, hybrid learning, machine learning, mechanism of action, peptide therapeutics

## Abstract

Cancer is a group of diseases causing abnormal cell growth, altering the genome, and invading or spreading to other parts of the body. Among therapeutic peptide drugs, anticancer peptides (ACPs) have been considered to target and kill cancer cells because cancer cells have unique characteristics such as a high negative charge and abundance of microvilli in the cell membrane when compared to a normal cell. ACPs have several advantages, such as high specificity, cost-effectiveness, low immunogenicity, minimal toxicity, and high tolerance under normal physiological conditions. However, the development and identification of ACPs are time-consuming and expensive in traditional wet-lab-based approaches. Thus, the application of artificial intelligence on the approaches can save time and reduce the cost to identify candidate ACPs. Recently, machine learning (ML), deep learning (DL), and hybrid learning (ML combined DL) have emerged into the development of ACPs without experimental analysis, owing to advances in computer power and big data from the power system. Additionally, we suggest that combination therapy with classical approaches and ACPs might be one of the impactful approaches to increase the efficiency of cancer therapy.

## 1. Introduction

Cancer is caused by genetic mutations [1] and has six distinct characteristics, including sustaining proliferative signaling, evading growth suppressors, resisting cell death, enabling replicative immortality, inducing angiogenesis, and activating invasion and metastasis [2]. It is the major leading cause of death worldwide, and it is a significant burden on society [3]. The traditional approaches for cancer therapy including chemotherapy and radiation therapy may lead to serious side effects [4]. For instance, it has been reported that cancer cells develop resistance toward anticancer drugs [5]. Radiation therapy has some side effects, such as sore skin, fatigue, hair loss, and problems with eating and drinking [6,7,8]. Another approach, immunotherapy, showed low efficacy, only 10~30% [9]. So far, overcoming limitations with conventional anticancer therapy is a huge challenge. Hence, it is necessary to develop novel therapeutic anticancer drugs.

The concept of peptide therapeutics in the field of medicine was introduced in 1922 for type 1 diabetes with insulin extracted from animal pancreases [10]. Du Vigneaud’s group firstly used chemically synthesized polypeptides of oxytocin and vasopressin, which are pituitary neuropeptides [11]. Since the peptide therapeutics are efficacious, relatively safe, highly selective, well-tolerated, have less side effects, and have low production costs [12,13,14,15], these are used in pharmaceutical research and development and clinical trials for diabetes, osteoporosis, HIV infection, chronic pain, and cancer [12,16,17].

Among the peptide therapeutics, anticancer peptides (ACPs) are of great interest due to their characteristics of selective and therapeutic properties toward cancer cells [18,19]. However, the identification of ACPs through wet-lab experimentation is time-consuming and expensive. Additionally, short half-life caused by peptidases, unknown immunogenicity, and low oral bioavailability are limitations for the use of ACPs [20]. Nevertheless, ACPs can be promising candidates for anticancer therapeutics owing to their high selectivity, high penetration, few side effects, and ease of chemical modification [18,21]. Additionally, multiple peptide-based therapies against various cancers have been investigated and are being developed in various phases of preclinical treatment and clinical trials [22,23], confirming the need for developing novel ACPs for cancer treatment. Therefore, the development of an efficient computational method is essential to identify potential ACP candidates before in vitro tests. To this end, computational methods such as machine learning (ML) and deep learning (DL), which are a subset of artificial intelligence (AI), have been developed to facilitate high-throughput screening of ACPs [21,24,25].

Some cancer treatments do not work well with single approaches due to the complex and heterogeneous characteristics of cancer cells. So far, single cancer therapeutic approaches remain a limitation in cancer therapy [26]. Thus, the combination of classical therapy with the ACP strategy can be a potential therapy to increase efficiency. In this review, we mainly focus on the cutting-edge AI for ACP prediction and cover the mechanism of ACPs in cancer in the following three sections: (i) development of therapeutic ACPs; (ii) application of ML and DL for ACP development; and (iii) future approaches of combinational therapy with ACPs for cancer therapeutics.

## 2. Development of Therapeutic ACPs

Peptides have been accurately synthesized by various methods in solid phase or solution [27]. Moreover, peptides have been used as therapeutics for cancer, hormone regulators, antibiotics, inflammation modulators, vaccines, drug-delivery systems, quorum sensing molecules, and so on [24,28,29,30,31,32]. In addition, peptides are also used as drug carriers by cell-penetrating and cell surface-binding properties of peptides [33,34,35]. Recently, it has been shown that more than 600 peptidic compounds were examined in the preclinical or clinical trials and over 60 *peptide* drugs were approved in the world market [17,36]. Novel ACPs are being steadily discovered, and several ACPs are approved by the FDA and EMA (Table 1) [37,38].

The development and identification of peptide drugs have been performed in in vitro assays [39], computation-aided rational designs for peptide–protein interaction interfaces [40], and mass spectrometry-based identification [41]. These traditional methods have been often time-consuming and prohibitively expensive. Thus, a reduction of the costs and rapid screening of new ACPs have become an urgent need in the pharmaceutical industry. The development of sequence-based computational methods using AI is helpful to identify ACP candidates before their validations in in vitro assays. Thus, the development of sequence-based computational methods, especially ML, allows for the rapid prediction of therapeutic peptides, because ML creates a back-and-forth selection process and dramatically reduces the time needed for selecting optimal sequences by the analysis of previous experimental peptide sequences, and allows the rapid identification of potential peptide drug candidates prior to their experimental validations [42,43,44]. Additionally, DL is also applied to predict ACPs by extracting features from peptide sequences [45,46].

Computational algorithms with support vector machines (SVM) have been applied using data sets of approximately 100–200 samples for ACP prediction by Tyagi et al. (AntiCP) [47] and Hajisharifi et al. [48] in early development. So far, a lot of ACP predictors using ML have been developed such as ACPP [49], iACP [50], iACP-GAEnsC [51], MLACP [24], SAP [52], ACPred-FL [53], mACPpred [54], ACPred [55], PEPred-Suite [56], DRACP [57], and AntiCP 2.0 [46]. Especially, ML models predict how peptide sequences affect target cells or diseases without physical and biological analyses, owing to advances in computer power, algorithm power, and big data from power systems [58,59,60,61]. DL is also used for ACP development in ACP-DL [45], PTPD [62], DeepACP [63], and ACPred-LAF [64]. Especially, in DeepACP, CNN, RNN, and CNN-RNN models were compared, and RNN showed the best performance [63]. Additionally, hybrid learning is used for ACP development in ACP-DA [65] and by Lv et al. [25]. AI tools for ACP prediction are summarized in Table 2. Therefore, computational methods with ML and DL allow the identification of new potential ACPs and are cheaper, more effective, and quicker than traditional methods. We divide this section into three subsections: classifications of ACPs, characteristics of ACPs, and therapeutic mechanisms of ACPs in cancer cells.

### 2.1. Classifications of ACPs

ACPs can be classified by structural properties of the peptides, such as α-helical, β-pleated sheet, random coil, and cyclic ACPs (Figure 1A) [66]. For instance, hydrophobic residues of ACPs enhance the cationic properties of α-helical structures, and the amphipathic properties of α-helical structures play an important role in cytotoxicity for cancer cells [55]. Disulfide bridge formation in β-pleated sheets is essential for structural maintenance. Generally, β-pleated ACPs have lower anticancer activity than α-helical ACPs, and their toxicity to the normal tissue is also lower [66].

ACPs can be classified by their actions; (1) those that directly act on cancer cells with cytotoxic activity by molecular interaction; (2) those that indirectly act on cancer cells with immune cell-stimulating activity to kill cancer cells (Figure 1B) [18]. The directly acting peptides usually bind to specific or overexpressed molecules in cancer cells, and peptides derived from defensins, lactoferricin B, cecropins, magainin-2, and chrysophsin-1 are included in this group [67]. Immune cells stimulating peptides is an alternative anti-cancer approach that uses the host’s immune system [68]. The peptides are used as T-cell antigens, also called peptide cancer vaccine [69,70], and recruit activated natural killer cells to cancer cells [71].

ACPs can be also classified according to the strategy for obtaining peptides: (a) natural ACPs derived from the natural peptides of plants, animals, and humans; and (b) modified ACPs using recombinant technology and chemical synthesis (Figure 1C). Natural ACPs occur in nature in the form of fragmented proteins from plants, animals, and humans [72]. These peptides can act as potent agonists and antagonists for molecules associated with disease progression [73]. Cathelicidin is an example of ACPs exerting a membranolytic activity against cancer cells [74]. Among the 30 Cathelicidin family members in mammals, only hCAP-18 has been identified in humans from neutrophils, monocytes, and mast and dendritic cells [75]. When hCAP-18 is cleaved by serine proteases, leucine-leucine-37 (LL-37) is produced and has been reported to be involved in adaptive immunity, growth inhibition, chemotaxis, and wound healing, and specifically induces destabilization of the cancer cell membrane by toroidal pore mechanism [74]. Although several studies indicated that LL-37 has a dual role as a cancer suppressor and oncogene, LL-37 still has the potential to be an anticancer agent [76]. Another example is human defensins, peptides produced by neutrophils and epithelial cells [77]. Among defensins, HNP-1, HNP-2, and HNP-3, named α-defensins, have been reported to have cytolytic activity and can induce apoptosis by either an extrinsic or an intrinsic pathway in cancer cells [74]. Human β-defensin-3 (hBD3) also performs anticancer activity by directly binding to phosphatidylinositol 4,5-bisphosphate (PI(4,5)P_2_) on the cell membrane and mediating cytolysis [78]. Although natural ACPs have a beneficial effect on cancer treatment, susceptibility to proteolysis has been problematic for the application of these peptides [79]. Hence, it leads to a pressing need to design optimal peptides and perform modifications of ACPs to solve this problem.

Replacing original amino acids with unnatural ones can be one of the strategies for both enhancing effectiveness against target cells and resisting proteolytic degradation [80]. Melittin (MEL), a 26-amino acids peptide, showing strong inhibitory effects on prostate, lung, liver, and ovarian cancer cells, can interact with negatively charged phospholipids on the cancer cell membrane. When valine in the 8th site and proline in the 14th site were replaced by lysine, it was able to inhibit the growth of BEL-7402/5-FU cells in mice and be cytotoxic on cancer cells and not on normal cells [81].

D-amino acids, which are enantiomers of natural L-amino acids with the same chemical and physical properties [82], can be one of the breakthroughs for improving the effectiveness of ACPs. Unlike L-amino acids, D-amino acids were not easily degraded by endogenous proteases in vivo [83]. Thus, the replacement of L-amino acids with D-amino acids leads to increased serum stability [84]. In addition, in several cases, enhanced capacity to kill cancer cells by D-amino acid analogs of peptides has been reported [85,86]. For instance, D-K6L9 is an engineered ACP, consisting of only lysine and leucine amino acids, and its natural amino acid sequences have been substituted with D-amino acids [74,87]. The K6L9 with D-enantiomer showed higher stability than K6L9 with L-amino acids and higher effectiveness in the reduction of prostate cancer size by inhibiting the secretion of prostate-specific antigen in serum [88]. Modified PMI (TSFAEYWNLLSP) increased the anticancer activity with high stability and protease resistance [89,90]. Hence, a simple modification of peptides can lead to enhanced anticancer properties and improve the efficacy of cancer therapy.

Tumor-homing peptides target molecules that are specifically overexpressed in cancer cell membranes and cancer-associated endothelial cells [91]. Some tumor-homing peptides bind the molecules and activate or inhibit cell signaling, including cell death, proliferation, and cellular activity [91]. Among the tumor-homing peptides, iRGD (CRGDKGPDC), which contains integrin binding motif (RGD) and C-end Rule motif (R/KXXR/K, activated by proteolytic cleavage), has been used for tumor-homing peptide and drug delivery studies [92,93,94]. The RGD motif-containing peptides preferentially bind to α_v_β_3_ integrin, which is preferentially expressed in cancer cells at specific stages, and tumor blood vessels [95,96]. The α_v_β_3_ integrin antagonist effect of the RGD peptides exhibits an anti-tumor effect by anti-angiogenesis and inhibition of tumor growth [96]. In addition, the C-end Rule motif can be uncovered by proteolytic cleavage after tumor-homing, the uncovered R/KXXR/K motif can bind to neuropilin-1 (NRP1), and the NRP1 binding activates the endocytic bulk transport pathway and increases tumor tissue permeability [95]. Thus, iRGD has been used as an ACP and drug delivery tool in various tumor types [97].

### 2.2. Characteristics of ACPs and Therapeutic Mechanisms of Cationic ACPs in Cancer Cells

The activities of ACPs are affected by the composition of amino acids and their structure [18]. In the reported ACP databases (from ACP-DL, ACPP, ACPred-FL, AntiCP, iACP, CancerPPD, APD3, and SATPdb) [98,99,100], positively charged or hydrophobic amino acids (Gly, Ala, Phe, His, Lys, Leu, and Trp) have been highly observed in ACPs, compared to non-ACPs [46,101]. Especially, the positively charged amino acid, Lys, was frequently observed in ACPs, whereas the frequencies of negatively charged amino acids, Asp and Glu, were less than other amino acids in 1390 ACPs database analysis [101]. The reported AI tools and database for prediction are summarized in Table 2 and Table 3.

Cancer cell surfaces are negatively charged because of the increase in the expression of anionic molecules such as phosphatidylserine, O-glycosylated mucins, negatively charged gangliosides, and heparan sulfates and their exposure [102,103]. In addition, one feature of cancer is extracellular acidification (Figure 2) [104]. It is induced by lactate secretion from increased glycolysis [105], and proton secretion by transporters and pumps such as sodium–hydrogen exchanger, monocarboxylate transporter, and V-ATPase [104]. The blood supply is limited in cancer tissues, so the oxygen concentration is lower than that of normal tissues [106,107]. In these hypoxic conditions, cancer cells increase the expression of carbonic anhydrase IX (CAIX), which can reversibly catalyze the carbon dioxide to bicarbonate and proton, contributing to an acidic environment [108,109]. Consequently, the extracellular pH (pHe) of cancer cells is maintained lower (pH 6.2–6.9) than that of normal cells (pH 7.3–7.4) [103]. The relatively larger number of microvilli in cancer cell membranes is another difference that distinguishes them from normal cells, which increases the surface area of the cells [18,102,110]. In contrast to the feature of the cancer membrane, ACPs have a positive net charge generally [18,66]. Moreover, as the pH of the microenvironment is lowered, the net charge of the protein becomes more positive. Therefore, cationic ACPs selectively interact with cancer cells and penetrate the membrane [18,111]. Cationic ACPs exert cytotoxic effects on cancer cells through various kinds of mechanisms (Figure 3). Cationic ACPs damage the cancer cells through apoptosis and necrosis by disrupting the membrane integrity [18]. Cationic ACPs are also internalized into cells and interact with several intracellular proteins and exhibit anticancer effects [112,113]. These cationic ACPs inhibit the activity and action of proteins (kinases, proteases, or other functional proteins) by interfering with protein–protein interactions directly or by modulating their conformational changes [113]. For example, the cationic anticancer peptide called RT53, which mimics the heptad leucine repeat of AAC-11, has a selective cytotoxic effect on cancer cells by inhibiting the anti-apoptotic properties of AAC-11 [113,114]. In the case of cell-internalized cationic ACPs, they induce the cytochrome c (Cyt c) release by disruption of the mitochondrial membrane and induce mitochondrial-dependent apoptosis [115]. It has also been found that cationic ACPs suppress angiogenesis by interfering with interactions between growth factors and their receptors [112]. On the other hand, some cationic ACPs perform the immunomodulatory function by increasing cytokine secretion, recruiting leukocytes, or activating immune cells [66,112,116]. For example, bovine lactoferrin (LfcinB) can alter cytokine production and enhance host defense against cancer [66,117]. The levels of proinflammatory cytokines including IL-6, IL-8, TNFα, and GM-CSF were inhibited in lactoferrin-treated murine squamous cell carcinoma cell line (SCCVII), and cancer growth was delayed in lactoferrin-treated mice [118]. In addition, alloferons, naturally occurring biological molecules primarily derived from insects, can stimulate natural killer cells (NK cells) and induce cytotoxicity in cancer cells through the stimulated NK cells [112,119].

## 3. Application of ML and DL for ACP Development

Novel drug discovery is challenging and takes a lot of time and money [120]. The processes of new drug discovery can be divided in to four stages: (1) target selection and validation, (2) screening and optimization of compound, (3) preclinical studies, and (4) clinical trials. After all in vitro and in vivo examinations, the drug candidate is reviewed for approval and commercialized by FDA [121]. This traditional workflow takes over 12 years and the cost has been estimated to be around $2.6 million [122]. Hence, the way to reduce the costs and accelerate the development of a candidate is a common interest.

Along with the advancement of technologies and flooding digital data of pharmaceutical sectors, AI enables managing a large number of data and is diversely applied in the pharmaceutical field [123]. AI in chemical-based drug development is useful for primary and secondary drug screening [121] and predicting drug–target interaction [124]. In addition, predictions of pharmacological properties [125]; potential efficacy [126]; and in silico absorption, distribution, metabolism, excretion, and toxicity (ADMET) [127] of a drug candidate became a reality with computational approaches. These active involvements of AI are expected to make the development of new drugs quicker and more cost-effective.

The overall process of using AI methods is to input the data on ACPs and non-ACPs and perform feature extraction, classification, and prediction. Before applying AI methods, it is necessary to split the data of ACPs and non-ACPs. Data split is divided into train, test, and validation sets to evaluate and test the model [128]. AI methods have been applied to the development, identification, and prediction of ACPs using ML [46], DL [45], and hybrid learning [25] (Figure 4).

In the case of ML, peptide features are extracted directly by a researcher. The feature extraction improves the prediction accuracy of the model by removing unnecessary and irrelevant features [129]. The feature extraction is performed by extracting relevant features [130], measuring the importance values of the features [131], and reducing their dimensions [132]. After the feature extraction, classification is conducted with the ML models. Among these ML models, SVM [133] is used the most with different feature extraction values to predict ACPs, and other ML models include *k*-nearest neighbor (*K*NN) [134], random forest (RF) [135], ensemble [136], and light gradient boosting machine (LightGBM) [137]. SVM takes the strategy of obtaining the maximum margin hyperplane [133]. KNN classifies which group each of the existing data groups belongs to when new data comes in [134]. RF is a classifier based on a decision tree using bootstrap sampling and random feature selection [135]. The ensemble uses multiple ML models to integrate the results of the models and make a final prediction [136]. LightGBM is one of the gradient-boosting models using the leaf-wise tree growth method [137]. In the case of DL, a data split is performed in all models and then the data embedding process is undertaken. Embedding quantizes peptide sequence data into a matrix [25]. The embedding matrix is used to perform feature extraction with DL models. Long short-term memory (LSTM) [25], convolutional neural network (CNN) [63], CNN-recurrent neural network (RNN) [63], and attention [64] structures are used as DL models. CNN is a neural network architecture for DL that learns directly from data [138]. RNN is a sequence model that processes inputs and outputs in sequence units [139]. Among the RNN models, LSTM has a memory cell; therefore, long sequences can be learned [140,141]. Attention is a model that intensively learns important parts using an encoder-decoder structure [142]. After feature extraction with DL, classification is performed through dense and sigmoid layers [45,143]. Hybrid learning is a method of combining DL and ML. In Lv et al.’s hybrid learning process, data split, embedding, and feature extraction are performed by the DL method, and classification is performed by the ML method [25]. On the other hand, in ACP-DA, data split and feature extraction are performed by the ML method, and classification is performed by the DL method [65]. After classification in AI methods, prediction is performed [144]. The validation set is used to verify the model and select the model with the best performance among the models. Finally, the prediction is evaluated with the test set [145,146].

Despite the large amount of effort put into the prediction of ACPs using AI, ACPs for the treatment of human cancer have not yet been explored well. However, some research reports that peptides developed by AI have anticancer activity on cancerous cell lines and negligible toxicity on normal cell lines (Appendix A) [147,148,149,150]. Hence, it is true that the efficacy and safety of ACPs developed by AI are not well explored yet, but they have high potential to be a promising candidate for cancer therapy and can be applied for preclinical and clinical trials later.

ACPs predicted by AI must be subjected to a complex evaluation process that includes biological functional validation, optimization, preclinical studies, and clinical trials. In the first step, it takes a long time and is expensive to do biological functional validation for ACPs. Thus, decreasing costs and fast process are the main requirements in biological functional validation. To this end, methods for predicting various biological functions with AI have been developed in peptide therapeutic drugs. These include the use of AI in biological functional validations such as anti-inflammatory, developed using data obtained from the IEDB database [151,152,153] and AIPpred [31]; proinflammatory (PIP-EL) [43]; cell-penetrating (CPPsite/CPPsite 2.0) [154,155]; anti-hypertensive (AHTPDB [156], mAHTPred [157], and BIOPEP [158]); B-cell epitope prediction (iBCE-EL) [35]; and hemolytic (HLPpred-Fuse) activities [159]. Therefore, it would be better to develop ACPs using AI in the future and analyze the various biological functions above first and then analyze the anticancer activity by in vitro experimental validation.

## 4. Future Approaches of Combinational Therapy with ACPs for Cancer Therapeutics

So far, overcoming the limitations of conventional anticancer therapy remains a challenge. Combination therapy with traditional therapy and ACPs is a promising therapeutic strategy for various kinds of cancers (Figure 5).

Cancer treatment using chemotherapy has been widely conducted owing to its high survival rate [160]. Chemotherapeutic drugs, including cyclophosphamide and cisplatin [161], can affect the rapid proliferation of cancer cells [162]. However, despite its great efficacy in killing cancer cells, chemotherapy has shown several side effects by inducing toxicity on healthy normal cells [163], and cancer resistance against drugs has also increased [164]. Cisplatin is a frequently used chemotherapeutic drug for advanced gastric cancer, but its toxicity is generally increased in the case of addition to single agents or chemotherapy doublets [165]. In comparison with a full dosage of cisplatin alone, co-administration of cisplatin with anticancer bioactive peptide-L (ACBP-L), isolated from goat liver, enabled the reduction of cisplatin dose from 5 mg/kg every 5 days to 5 mg/kg every 10 days with efficacy and improved the quality of life in a xenograft nude mouse model bearing MGC-803 in vivo [166]. Thus, combinational therapy with chemotherapy and ACPs could reduce the burden for patients to endure in the future.

The regulation of mitochondrial metabolism is a target for cancer therapy in chemotherapeutic drugs. As mitochondria play a pivotal role for oncogenesis [167], the application of materials related to mitochondrial metabolism, such as glutamate dehydrogenase inhibitor, in combination with anticancer therapies, may be able to enhance anticancer effects [168]. Among mechanisms through which ACPs can affect cancer cells, one of the leading systems is the induction of mitochondrial dysfunction and programmed cell death or apoptosis [169]. Thus, the administration of mitochondrial-targeting ACPs can be useful for cancer therapy. For this reason, we suppose that combinational therapy with ACPs and mitochondrial-targeting drugs for metabolic reprogramming can increase the therapeutic efficiency in cancer cells, as metabolic reprogramming in cancer cells is different from that in normal cells, owing to genetic alteration and differences in nutrient and oxygen availability [170,171]. Moreover, metabolic reprogramming is highly related to resistance against cancer therapy [172]. Among metabolic reprogramming, it has been reported that most of the cancer cells show increased glutaminolysis, which is enzymatic conversions of glutamine to α-ketoglutarate for the generation of ATP in mitochondria [173,174]. Thus, anticancer approaches have been trying to block glutaminolysis by targeting the glutamine transporters and enzymes, including glutaminase (GLS), glutamate dehydrogenase (GDH), glutamate pyruvate transaminase (GPT), and glutamate oxaloacetate transaminases (GOT) [175]. In the case of glutamine transporters, alanine-serine-cysteine transporter 2 (ASCT2, also known as SLC1A5) and L-type amino acid transporter 1 (LAT1, also known as SLC7A5) are overexpressed in most cancers [176,177,178]; 2-aminobicyclo(2,2,1)-heptane-2-carboxylic acid (BCH) inhibits the function of LAT1 [179]; γ-L-glutamyl-p-nitroanilide (GPNA) inhibits the function of ASCT2 [180,181]; and benzylserine inhibits the function of LAT1 and ASCT2 [182]. In the case of inhibition of GLS activity, bis-2-[5-phenylacetamido-1, 2, 4-thiadiazol-2-yl] ethyl sulfide (BPTES) [183], CB-839 [184], and compound 968 [185] are reported as blockers of the GLS. In the case of inhibition of GDH activity, epigallocatechin gallate (EGCG), purpurin, and R162 are reported as blockers of the GDH [186,187]. Aminooxyacetate (AOA) is reported as a blocker of GPT and GOT activities [188,189]. These drugs for the inhibition of glutaminolysis have been validated for various kinds of cancers. Therefore, the combination therapy with the targeting of glutaminolysis drugs and ACPs might be one of the approaches to increase the efficiency of cancer therapy.

Immunotherapy, including ipilimumab and nivolumab, is primarily aimed at strengthening the immune system so that immune cells identify and eliminate cancer cells [190,191]. As therapeutic approaches for the immune system, including immune checkpoint inhibitors (ICIs), adoptive cell therapy, oncolytic viruses, and cancer vaccines have been developed, and their clinical applications are expanded [192]. However, some mutated cancer cells are less antigenic and, hence, escape from immune effect; this leads to the low efficacy of immunotherapy [9]. The administration of CpG oligodeoxynucleotides (CpG-ODN) could be a promising method to control ovarian cancers by targeting Toll-like receptor 9 (TLR9) and activating the immune system [75]. Co-administration of LL-37, cleaved form of 18 kDa human cathelicidin protein (hCAP18), with CpG-ODN, generates synergistic effects on anticancer activity and increases survival in MOSEC/luc cancer-bearing mice compared with respective treatment with each drug [193]. Hence, co-treatment of therapeutic peptides for cancer with immunotherapy can enhance the efficacy of cancer therapy.

Radiotherapy is also a traditional cancer therapy, delivering high-energy photons and making secondary electrons in human tissues, which can cause DNA damage, leading to the impairment of cell division [194]. For instance, in radiotherapy, radioactive iodine is widely used against thyroid cancer [195]. However, this method also has some side effects, such as sore skin, tiredness, hair loss, and problems with eating and drinking [6,7,8]. It has also been reported that a combination of ACPs and radiotherapy could exert a synergistic effect on killing cancer cells [196]. The oncolytic peptide LTX-315, a chemically modified 9-mer cationic peptide, is a highly effective ACP that induces immunogenic cell death in cancer cells [197]. In addition, the effect of LTX-315 on breast cancer via activation of anticancer immunity can be boosted by radiation therapy [196]. This result strongly suggested that combinational therapy with radiotherapy and ACPs might increase the therapeutic efficiency for cancer cells.

Hyperthermia, also known as thermotherapy, is conducted by the elevation of body temperature using electromagnetic radiation [198,199]. High body temperatures above 41–42 °C can kill cancerous cells by affecting membranes, cytoskeleton, synthesis of proteins, and DNA repair [200]. However, hyperthermia effectiveness is low as a single treatment, and malignant and non-malignant cells are sensitive to heating in general; therefore, it is scarcely included in modern oncological management [201]. Thermal targeting by the addition of KLAKLAKKLAKLAK (KLAK) peptide to the C-terminus of the heat-responsive biopolymer elastin-like polypeptide (ELP) and increasing the penetration ability into cells by the addition of cell-penetrating peptide sequence (SynB1) to the amino terminus of ELP, KLAKLAKKLAKLAK (KLAK) peptide showed improvement in targeting cancer cells with the application of mild hyperthermia [202]. This report suggests that the therapeutic effects of thermotherapy depend on the peptide sequence, which should be considered in thermotherapy.

Hormone therapy, also called endocrine therapy, has been utilized for menopausal symptoms and breast cancer. Cancers, including breast cancer and prostate cancer, which are controlled by reproductive hormones, are targeted, and apoptosis is induced by extrinsic hormone receptor-mediated death pathways with the administration of exogenous hormones, such as estrogen, progestin, and luteinizing hormone-releasing hormone (LH-RH) [203,204]. Goserelin and Leuprolide are commonly used for prostate cancer as gonadotropin-releasing hormone (GnRH) receptor agonists [205]. However, as some hormone analogs have shown severe sexual dysfunction and suppressed ovarian function, hormone therapy should be carefully applied for cancer treatment [206]. FK506-binding protein-like (FKBPL)-based therapeutics, AD-01 and ALM201, showed toxicity against cancer stem cells (CSCs), which are resistant to endocrine therapy. The combination of ALM201 with tamoxifen, frequently used as an endocrine therapy drug for breast cancer, effectively delayed cancer recurrence by significantly reducing the number of mammospheres formed by tamoxifen-resistant CSCs in ER+ MCF-7 xenografts, compared to a single treatment of tamoxifen and ALM201 [207]. Thus, this report suggests that the combination with hormone therapy and the development of peptide drugs for cancer therapy are needed.

Photodynamic therapy, using photosensitizing agents such as photofrin and foscan, activates the agents with the light of a specific wavelength and leads to severe photodamage to cancer cells [208,209]. It is widely used for many types of cancers; however, photodynamic effects only occur at the irradiated place, not allowing whole-body irradiation, and deep cancers without surgery are hard to be eliminated, owing to low tissue penetration of light [210]. When D-(KLAKLAK)_2_ peptide, a cytotoxic peptide that disrupts mitochondrial membranes in cancer cells [211], is conjugated to photosensitizer protoporphyrin (PpIX) with PEG linker, the presence of D-(KLAKLAK)_2_-induced cytotoxic effect on HeLa cells at a relatively low dose of light irradiation enhances the efficacy of photodynamic therapy [212]. These reports suggest that the combination of photodynamic therapy and ACPs induces synergetic effects in cancer therapy.

## 5. Conclusions

ACPs have a lot of merit in activities for apoptotic, cell-penetrative, anti-inflammatory, and anti-angiogenetic effects in cancer cells both in vitro and in vivo. Although there are continuous evoked issues in ACP-related studies, there are strong positive outcomes in ACP-related research. Computational approaches with the application of ML, DL, and hybrid learning save time and cost for the identification of efficient ACP candidates before the wet-lab experiment. Moreover, before the experimental validation (including biological functional validation, optimization, preclinical studies, and clinical trials) of the therapeutic cancer effects of candidate ACPs predicted by AI, it is helpful to utilize AI to predict various biological functions of new ACPs. Additionally, single cancer therapeutic approaches have limitations for cancer therapy. Thus, the combination of classical therapy with the ACP strategy could be a potential therapy to increase efficiency. In conclusion, this review may provide a rationale for further research on the development of ACPs based on cancer cell characteristics and facilitate understanding AI and combinational therapy for cancer.

## Figures and Tables

**Figure 1 pharmaceutics-14-00997-f001:**
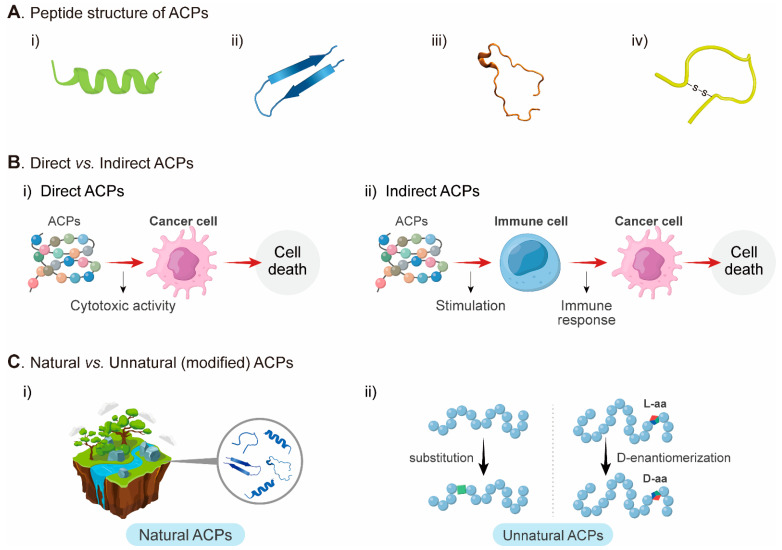
Classification of ACPs. (**A**) (i) α-helical; (ii) β-pleated sheets; (iii) random-coil; (iv) cyclic ACPs. (**B**) (i) direct ACPs; (ii) indirect ACPs. (**C**) (i) natural ACPs; (ii) unnatural (modified) ACPs. ACPs: anticancer peptides; aa: amino acid.

**Figure 2 pharmaceutics-14-00997-f002:**
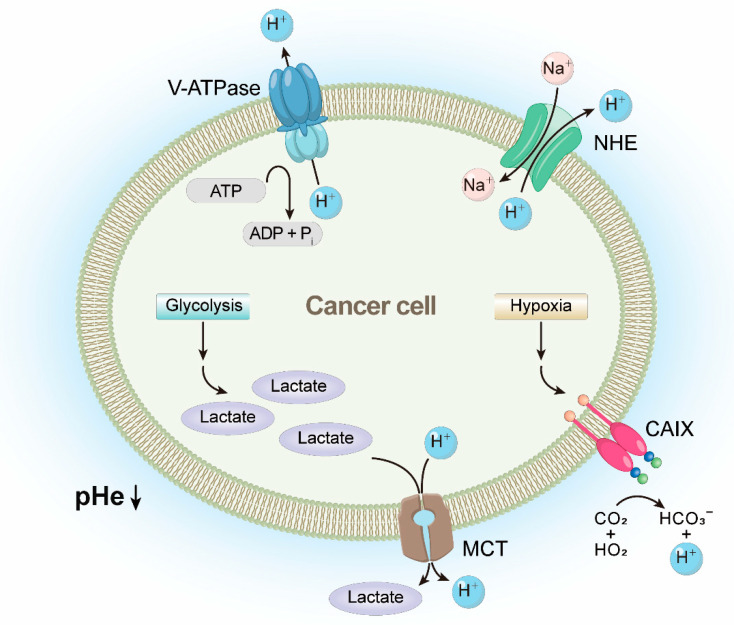
Extracellular acidification of the cancer cell. pHe: extracellular pH; NHE: sodium-hydrogen exchanger; MCT: monocarboxylate transporter; CAIX: carbonic anhydrase IX.

**Figure 3 pharmaceutics-14-00997-f003:**
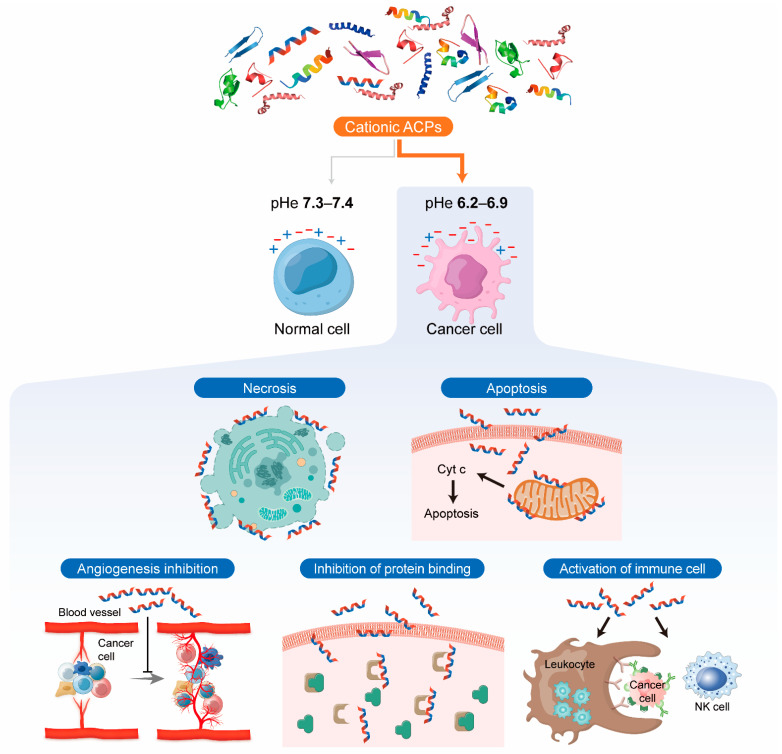
Therapeutic mechanisms of cationic ACPs in cancer cells. ACPs: anticancer peptides; pHe: extracellular pH; Cyt c: cytochrome c.

**Figure 4 pharmaceutics-14-00997-f004:**
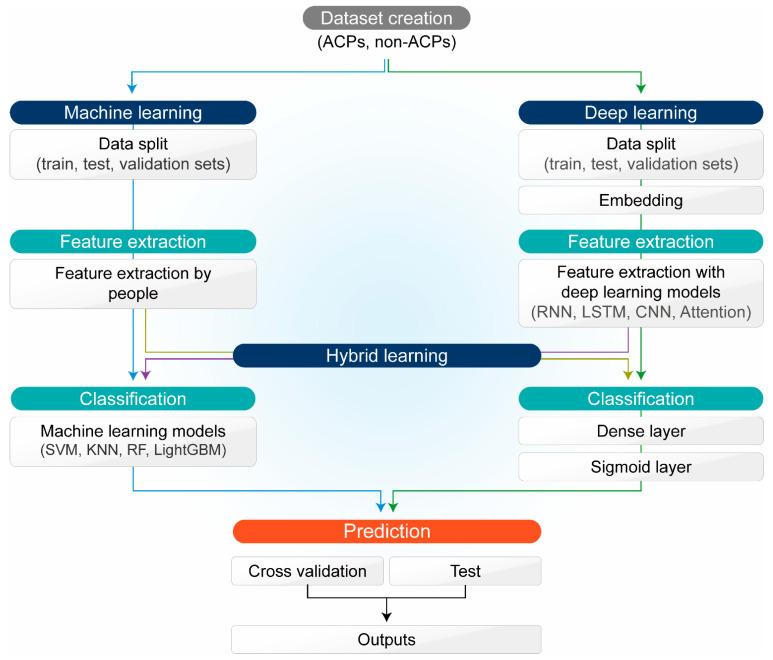
Schematic flowchart for development of ACPs using AI. RNN: recurrent neural network; SVM: support vector machine; *K*NN: *k*-nearest neighbor; RF: random forest; LightGBM: light gradient boosting machine.

**Figure 5 pharmaceutics-14-00997-f005:**
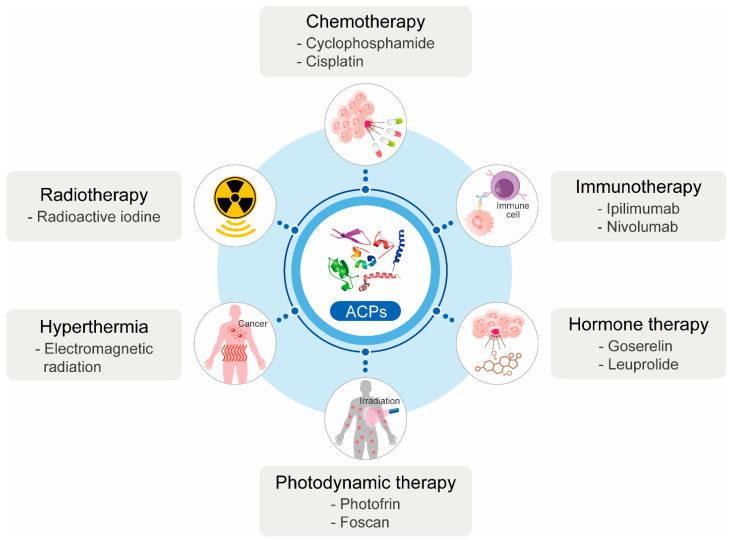
Combinational therapy of ACPs with other cancer therapies.

**Table 1 pharmaceutics-14-00997-t001:** List of ACPs approved by FDA and EMA.

Peptide Name	Brand Name	Indication	First Approval	Reference
Ixazomib	Ninlaro	Multiple myeloma	2015 (FDA)	[37]
Carfilzomib	Kyprolis	Multiple myeloma	2012 (FDA)	[38]
Bortezomib	Velcade	Multiple myeloma	2003 (FDA)	
Goserelin	Zoladex	Prostate cancer	1989 (FDA)	
Histrelin	Vantas	Prostate cancer	2004 (FDA)	
Leuprolide	Lupron	Prostate cancer	1985 (FDA)	
Degarelix	Firmagon	Prostate cancer	2008 (FDA)	
Romidepsin	Istodax	T-cell lymphoma	2009 (EMA)	
Thymalfasin	Zadaxin	Hepatocellular carcinoma	2002 (EMA)	
Triptorelin	Trelstar	Hormone-responsive cancers	2010 (FDA)	
Mifamurtide	Mepact	Osteosarcoma	2009 (EMA)	

ACPs: anticancer peptides, FDA: Food and Drug Administration (US), EMA: European Medicines Agency (EU).

**Table 2 pharmaceutics-14-00997-t002:** List of AI tools for ACP prediction.

Name	Datasets Size	Model URL	Method	Accuracy	References
MLACP	T: 187 ACPs and 398 non ACPs I: 422 ACPs and 422 non ACPs	http://www.thegleelab.org/MLACP/MLACP.html	ML	88.72%	[24]
Lv et al.	T: 861 ACPs and 861 non ACPs I: 970 ACPs and 970 non ACPs	https://github.com/zhibinlv/iACP-DRLF	Hybrid learning	93.5%	[25]
ACP-DL	T: 376 ACPs and 364 non ACPs I: 129 ACPs and 111 non ACPs	https://github.com/haichengyi/ACP-DL	DL	81.48% and 85.42%	[45]
AntiCP 2.0	T: 861 ACPs and 861 non ACPs I: 970 ACPs and 970 non ACPs	https://webs.iiitd.edu.in/raghava/anticp2/	ML	72.81% and 88.81%	[46]
Hajisharifi et al.	T: 138 ACPs and 206 non ACPs I: 22 ACPs	NA	ML	83.82% and 89.7%	[48]
ACPP	T: 217 ACPs and 3979 non ACPs I: 40 ACPs and 40 non ACPs	http://acpp.bicpu.edu.in/predict.php	ML	96%	[49]
iACP	T: 138 ACPs and 206 non ACPs I: 150 ACPs and 150 non ACPs	http://lin.uestc.edu.cn/server/iACP	ML	92.67%	[50]
iACP-GAEnsC	T: 138 ACPs and 206 non ACPs I: NA	NA	ML	96.45%	[51]
SAP	T: 138 ACPs and 206 non ACPs I: NA	NA	ML	91%	[52]
ACPred-FL	T: 250 ACPs and 250 non ACPs I: 82 ACPs and 82 non ACPs	http://server.malab.cn/ACPred-FL	ML	91.4%	[53]
mACPpred	T: 266 ACPs and 266 non ACPs I: 157 ACPs and 157 non ACPs	http://thegleelab.org/mACPpred/	ML	91.7%	[54]
ACPred	T: 138 ACPs and 205 non ACPs I: 250 ACPs and 250 non ACPs	http://codes.bio/acpred/	ML	92.87%	[55]
PEPred-Suite	T: 250 ACPs and 250 non ACPs I: 82 ACPs and 82 non ACPs	http://server.malab.cn/PEPred-Suite	ML	NA	[56]
DRACP	T: 138 ACPs and 206 non ACPs I: 150 ACPs and 150 non ACPs	https://github.com/zty2009/ACP	ML	96%	[57]
PTPD	T: 225 ACPs and 2250 non ACPs I: 138 ACPs and 206 non ACPs	NA	DL	96%	[62]
DeepACP	T: 250 ACPs and 250 non ACPs I: 82 ACPs and 82 non ACPs	https://github.com/jingry/autoBioSeqpy/tree/master/examples/anticancer_peptide_prediction	DL	84.9%	[63]
ACPred-LAF	T: 558 ACPs and 558 non ACPs I: 148 ACPs and 148 non ACPs	https://github.com/TearsWaiting/ACPred-LAF	DL	81.15%	[64]
ACP-DA	T: 376 ACPs and 364 non ACPs I: 129 ACPs and 111 non ACPs	https://github.com/chenxgscuec/ACPDA	Hybrid learning	82.03% and 88.33%	[65]

The database sets are accessed on 1 April 2022. T: training, I: independent, NA: not available, AI: artificial intelligence, ACP: anticancer peptide, ML: machine learning, DL: deep learning.

**Table 3 pharmaceutics-14-00997-t003:** List of ACP database.

Database	Total ACPs	Database URL	Reference
CancerPPD	3491	http://crdd.osdd.net/raghava/cancerppd/	[98]
APD3	185	http://aps.unmc.edu/AP/	[99]
SATPdb	1099	http://crdd.osdd.net/raghava/satpdb/	[100]

The database sets are accessed on 1 April 2022. ACP: Anticancer peptide.

## Data Availability

The data supporting the findings of this study are available from the corresponding author upon reasonable request.

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
