# Peer review of "Development of Anticancer Peptides Using Artificial Intelligence and Combinational Therapy for Cancer Therapeutics"

_pharmaceutics, 2022, doi:10.3390/pharmaceutics14050997_

Round 1

Reviewer 1 Report

The authors have tried to explain the concept of creating new anticancer peptides (ACPs) by adding artificial intelligence (AI) to the drug development pathway. Unfortunately, very little of the text is devoted to reflect the title of the article. Instead of this, the authors have devoted most of the manuscript to list different ACPs and their mechanism of action. The authors need to re-assess their text/title and re-consider what is the message they want to convey in their article. Is this a review article on ACPs or review article on how AI accelerates the drug discovery process of the ACPs as the title states?

This manuscript would be substantially helped if 1) the authors would be capable of expanding the perspective of their article outside of the field for readers not familiar with the field. To provide perspective, the authors should explain how many ACPs has been accepted by FDA for the treatment of human cancers and how much revenue they create/how many patients are being treated with these drugs. Furthermore, 2) they should clearly indicate those ACPs that were discovered/generated by AI assisted methods and made it to the clinical use in humans. Some kind of fact need to be provided that clearly demonstrate how much AI-based methods have accelerated drug development in the field of ACPs. 3) The article is hampered by limited understanding of the cancer biology in some parts of the manuscript, please see an example provided below:

  1. Acidic environment is caused by hypoxia in tumors. Carbonic anhydrases catalyze the reversible hydration of carbon dioxide to bicarbonate ions and protons, which causes acidic environment. Please revise lines 182 – 217 accordingly.
  2. As stated above, the manuscript would be strengthened by improved understanding of cancer biology.
  3. 1 is unacceptable. Excessive number of basic amino acids is not a feature of all ACPs. It is one strategy to obtain tumor-selectivity, but not a general requirement to be ACPs. The legend is also unacceptable in its present form.
  4. Tumor homing peptides that function as ACPs by delivering apoptosis inducing peptide (or being cytotoxic themselves, eg. iRGD) have been completely omitted from the manuscript. Please include this important group of peptides in the manuscript.
  5. More illustrations are needed to make the review article easier to read and follow than the current form.
  6. The article is a bit of list of different ACPs. I would write the manuscript by focusing more on general themes and limiting listing each individual ACP.

Reviewer 2 Report

In this article, the authors summarize the current status of anticancer peptides (ACPs), how to develop ACPs using AI, and the benefits of combining ACPs with conventional anticancer therapies with updates. The review will provide meaningful information for all readers, both researchers currently using ACPs and those considering using ACPs. If the following points are improved, the readers will be more benefited from this interesting article.

  1. It would be easier to understand the AI-assisted ACP predictors introduced in this review if authors could list their names, characteristics, and references in a table.
  2. It would be easier to understand if authors could list the names, characteristics, and references of the ACPs introduced in this review in a table.
  3. Please provide some examples of ACPs developed using ML/DL and their features and advantages.
  4. Please unify whether indentation is inserted at the beginning of a paragraph or not.
  5. Please unify whether a line is inserted between paragraphs or not.

Reviewer 3 Report

This very interesting review describes the development of anticancer peptides using artificial intelligence. It is well written and I definitely recommend to publish it after the minor revision.

I believe that the quality of this review will be greatly improved if the authors add the following structured table, which summarizes the research on anti-cancer peptides.

I think it is very important to add some spesific data on the efficacy in vivo of the described peptides developed by  artificial intelligence 

This table should contain the following fields:

  1. Peptide name
  2. Its amino acid sequence
  3. Method by which the peptide was developed, ML or DL or other technology
  4. Type of cancer tested in vivo: nosology, xenograft name, mouse or rat strain
  5. The maximum tumor growth inhibition index that was obtained with this peptide
  6. Relevant References

Reviewer 4 Report

The manuscript entitled “Development of anticancer peptides using artificial intelligence and combinational therapy for cancer therapeutics” submitted by Ji Su Hwang et al. reviews the benefits of Artificial Intelligence (AI) in anti-cancer peptides (ACPs) prediction and the related mechanism of ACPs in cancer.

This review provides an update on current knowledge and a useful document for large audience. Although this review presents an interesting approach, it remains that there are some minor points to elucidate:

  • In Figure 1, there is a redundancy of the scheme representing normal cell.
  • The authors should include a Table where they will indicate the different websites concerning the different methods of prediction. In this table, they should indicate the website, what we expect to predict and the related reference.

In conclusion, after the correction of these 2 points, I’ll recommend this manuscript for publication in "Pharmaceutics" journal.

Round 2

Reviewer 1 Report

The authors have revised the manuscript along the lines of my criticism and the manuscript has improved in the process. Furthermore, they have added very illustrative figures to the manuscript to enhance the readability.